# Comparative Chloroplast Genomics of *Litsea* Lam. (Lauraceae) and Its Phylogenetic Implications

Yunyan Zhang [1,†], Yongjing Tian [1,†], David Y. P. Tng [2], Jingbo Zhou [1], Yuntian Zhang [1], Zhengwei Wang [3], Pengfu Li [1,*] and Zhongsheng Wang [1]

[1] College of Life Sciences, Nanjing University, Nanjing 210023, China; zyynju@smail.nju.edu.cn (Y.Z.); mg1830075@smail.nju.edu.cn (Y.T.); mg1830084@smail.nju.edu.cn (J.Z.); 171850664@smail.nju.edu.cn (Y.Z.); wangzs@nju.edu.cn (Z.W.)

[2] Centre for Rainforest Studies, The School for Field Studies, Yungaburra, QLD 4884, Australia; davetngcom@gmail.com

[3] Shanghai Chenshan Botanical Garden, Shanghai 201602, China; wangzhengwei@csnbgsh.cn

* Correspondence: pengfuli@nju.edu.cn

† These authors contributed equally to this work.

**Abstract:** *Litsea* Lam. is an ecological and economic important genus of the "core Lauraceae" group in the Lauraceae. The few studies to date on the comparative chloroplast genomics and phylogenomics of *Litsea* have been conducted as part of other studies on the Lauraceae. Here, we sequenced the whole chloroplast genome sequence of *Litsea auriculata,* an endangered tree endemic to eastern China, and compared this with previously published chloroplast genome sequences of 11 other *Litsea* species. The chloroplast genomes of the 12 *Litsea* species ranged from 152,132 (*L. szemaois*) to 154,011 bp (*L. garrettii*) and exhibited a typical quadripartite structure with conserved genome arrangement and content, with length variations in the inverted repeat regions (IRs). No codon usage preferences were detected within the 30 codons used in the chloroplast genomes, indicating a conserved evolution model for the genus. Ten intergenic spacers (*psb*E–*pet*L, *trn*H–*psb*A, *pet*A–*psb*J, *ndh*F–*rpl*32, *ycf*4–*cem*A, *rpl*32–*trn*L, *ndh*G–*ndh*I, *psb*C–*trn*S, *trn*E–*trn*T, and *psb*M–*trn*D) and five protein coding genes (*ndh*D, *mat*K, *ccs*A, *ycf*1, and *ndh*F) were identified as divergence hotspot regions and DNA barcodes of *Litsea* species. In total, 876 chloroplast microsatellites were located within the 12 chloroplast genomes. Phylogenetic analyses conducted using the 51 additional complete chloroplast genomes of "core Lauraceae" species demonstrated that the 12 *Litsea* species grouped into four sub-clades within the *Laurus-Neolitsea* clade, and that *Litsea* is polyphyletic and closely related to the genera *Lindera* and *Laurus*. Our phylogeny strongly supported the monophyly of the following three clades (*Laurus–Neolitsea*, *Cinnamomum–Ocotea*, and *Machilus–Persea*) among the above investigated "core Lauraceae" species. Overall, our study highlighted the taxonomic utility of chloroplast genomes in *Litsea*, and the genetic markers identified here will facilitate future studies on the evolution, conservation, population genetics, and phylogeography of *L. auriculata* and other *Litsea* species.

**Keywords:** *Litsea auriculata*; core Lauraceae; comparative plastid genomics; microsatellites; DNA barcodes; phylogeny of *Litsea*

## 1. Introduction

The Lauraceae or the Laurel family is a large monophyletic family in the order Laurales and encompasses approximately 2500 to 3000 species from around 50 genera [1,2]. Systematic classification schemes and the tribe- and genus-level phylogenetic relationships within this family have long been controversial [1,3–6]. The focal genus of the present study, *Litsea* Lam., belongs to the "core Lauraceae" or "core Laureae" group in the sense of Chanderbali et al. [1] and Rohwer and Rudolph [7]. As currently circumscribed, *Litsea* contains around 408 species, with *Litsea cubeba* (Lour.) Pers. as its type species [8],

and this genus is commonly characterized by the following morphological traits: evergreen/deciduous trees or shrubs; leaves alternate, rarely opposite or verticillate; umbel inflorescence surrounded by an involucre consisted of persistent, alternate, and opposite bracts; anthers four-celled, all introrse, cells opening by lids [3,9]. The majority of *Litsea* species inhabit tropical and subtropical Asia, while a few *Litsea* species are distributed in Australia and from North America to subtropical South America [9]. Regarding the fossil records of *Litsea*, Dai et al. [10] reported a well-preserved fossil leaf of *Litsea* cf. *chunii* Cheng discovered in the Late Pliocene's sediments of the Mangbang formation in Tengchong county, Yunnan Province, China, and a mummified fossil wood of *Litseoxylon nanningensis* gen. et sp. Nov. was identified from the Upper Oligocene Yongning Formation of the Nanning Basin, Guangxi Province, South China, by Huang et al. [11]. Moreover, the origin and evolution of *Litsea* genera group sensu lato was discussed by Li [12], who pointed that this group originated at South Laurasia and North Gondwana, as well as the tropical coast of Tethys sea after Mid-Cretaceous; furthermore, the core genera of this group (*Litsea* and *Lindera*) probably originated and speciated ranging from South China to Indo-Malaysia, from where they migrated into tropical America and Australasia. Moreover, many of the species of *Litsea* have ethnobotanical or economic uses [13–18]. For instance, *Litsea cubeba* is an important source of May Chang essential oil (roughly 95% terpenoid), which is broadly used for perfumes and cosmetics [13–15], and *Litsea mollis* is used as a traditional Chinese medicine for the treatment of inflammation, poor blood circulation, lumbocrural pain, quadriplegia, and dysmenorrhea [18]. Additionally, many *Litsea* species are in urgent need of conservation: 98 species of this genus are listed in the International Union for Conservation of Nature (IUCN) Red List of Threatened Species (https://www.iucnredlist.org/ (accessed on 2 May 2021)) [19].

Despite the diversity and ecological and economic importance of *Litsea*, there are still uncertainties surrounding the systematic placement of the genus. Even though there have been many previous phylogenetic studies on *Litsea* that are based on morphological and anatomical characteristics as well as molecular data, systematic placement of *Litsea* remains perplexed. Richter [20] showed that *Litsea, Apollonias, Laurus, Lindera, Sassafras*, and *Umbellularia* shared similar wood structure. Li [21] found that *Litsea* and *Lindera* exhibited many similarities in terms of morphology and distribution. Specifically, Li [21] showed that the ancestral phenotype of the above two genera was characterized by umbellate flowers, elongated flower branches in epiphytic inflorescences, and evergreen leaves with three veins. Additionally, both genera originated from tropical Asia and followed identical evolutionary processes: from evergreen to deciduous, from three veins to pinnate veins, from more to less flowers, and from axillary to terminal inflorescences [21]. Li [21] also further suggested that *Litsea* and *Lindera* can be merged into a group if they did not display the variations in the number of anther cells (four-celled versus two-celled). In addition, Raj and Werff [22] made the case that *Litsea* was related to *Lindera* from the investigation of pollen morphology. Specifically, the size of pollen grains, the number of spinules per pollen grain, and the ultra-structural details of the pollen wall of *Litsea* and *Lindera* have been found to be identical [22]. On the basis of these morphological characters, Li [21] concluded that the number of anther cells could be a good taxonomic basis for the monophyly of *Litsea* within Lauraceae. However, Werff and Richter [23] discovered the homoplasy or instability of this character in distinguishing Lauraceae, thus resulting in the disputed systematics and delimitation of *Litsea*.

Rohwer [4] first used the universal chloroplast DNA gene *mat*K to construct a phylogenetic tree of Lauraceae, finding that the genera with umbellate, involucrate inflorescences (*Litsea, Actinodaphne, Laurus, Lindera, Neolitsea* and *Umbellularia*) had a low degree of *mat*K sequence differentiation and *Litsea* had a close relationship with *Lindera*. Chanderbali et al. [1] rebuilt the phylogenetic tree among 122 species of Lauraceae representing 44 genera employing the combined analysis of the chloroplast DNA genes (*trn*L–*trn*F, *trn*T–*trn*L, *psb*A–*trn*H, and *rpl*16) and ribosomal DNA sequences: 26S, internal transcribed spacer (ITS) and 5.8S, and their results indicated that *Litsea* grouped with *Actinodaphne, Lindera*, and

*Neolitsea*. Due to the unsettled generic delimitation of *Litsea* and *Lindera*, they cautioned against assessing the delimitations of these genera from morphology. On the basis of the *mat*K gene of chloroplast DNA and nuclear internal transcribed spacer (nrITS), Li et al. [3] analyzed the phylogenetic relationship of "core Laureae" group and found that *Litsea*, *Actinodaphne*, *Neolitsea*, and *Lindera* were polyphyletic but poorly bootstrap-supported. Fijridiyanto and Murakami [24] reconstructed the phylogenetic tree of *Litsea* and its related genera using the nuclear taxonomic marker *rpb*2, which supported the fact that *Litsea* was not monophyletic and closer to *Lindera*.

Over the last decade, the rapid advances of high-throughput sequencing or next-generation sequencing (NGS) technologies have yielded tremendous genome-scale data for angiosperm species and have improved the development of plastid phylogenomics and phylogenetic resolution of many land plants [25–27]. Sequencing complete chloroplast genome, for instance, has proven to be informative and effective in resolving complex phylogenetic relationships at a wide range of taxonomic levels [25–27]. Specific to the intrageneric phylogeny of the Lauraceae, these advances have revealed previously unknown phylogenetic relationships among genera. For example, Song et al. [6] employed the method of plastid phylogenomics to construct a comprehensive and robust phylogenetic tree of Lauraceae, finding that species of *Litsea*, *Actinodaphne*, *Iteadaphne*, *Laurus*, *Lindera*, *Neolitsea*, and *Parasassafras* grouped into one clade. Within the clade, four *Litsea* species (*L. glutinosa*, *L. monopetala*, *L. magnoliifolia*, and *L. tsilingensis*), *Laurus nobilis*, and four *Lindera* species (*L. communis*, *L. glauca*, *L. megaphylla*, and *L. nacusua*) were located in one sub-clade, while *Lindera obtusiloba* with other three *Litsea* species (*L. cubeba*, *L. panamonja*, and *L. pierrei*) formed another sub-clade. Tian et al. [28] investigated the phylogeny of the "core Lauraceae" group with their whole chloroplast genomes and found that three *Litsea* species (*L. monopetala*, *L. glutinosa*, and *L. tsinlingensis*), *Laurus nobilis*, and *Lindera megaphylla* formed one clade with high bootstrap. Using whole plastome genomes, Zhao et al. [29] also found strong support for a close relationship between the genera *Litsea*, *Laurus*, and *Lindera*, and suggested that the clade containing *Litsea, Laurus*, and *Lindera* was a sister to the "*Cinnamomum–Ocotea* clade" consisting of the genera *Cinnamomum*, *Nectandra*, and *Sassafras*. These studies have strongly indicated the polyphyletic nature of *Litsea* and their unresolved phylogenetic positions within the Lauraceae.

*Litsea auriculata* Chien et Cheng, the focal species of this study, is a rare deciduous tree and precious medicinal tree restricted and endemic to montane regions in the Zhejiang and Anhui provinces of eastern China [30,31]. Due to its narrow distribution and small number of wild populations, *L. auriculata* has been listed as "near threatened" by the IUCN and "Grade III Key Protected Wild Plant" by the Chinese Plant Red Book [32]. Thus far, research on *L. auriculata* has primarily focused on its seed germination, ecophysiology, and population genetics [30,33–35], while studies on the conservation genetics and genomics of the species is lacking. Therefore, we sequenced and reported the complete chloroplast genome of *Litsea auriculata* for the first time and conducted comparative chloroplast genomics of the genus *Litsea* with published chloroplast genomes of 11 other *Litsea* species. Moreover, we developed potential genetic markers such as microsatellites and DNA barcodes for the genus *Litsea*. Furthermore, to shed light on the systematic placement of *Litsea* within the "core Lauraceae" or "core Laureae" group, we reconstructed maximum likelihood (ML) and Bayesian inference (BI) phylogenetic trees of these *Litsea* species and an additional 51 published plastomes of "core Lauraceae" species. The outcomes here will provide useful genomic resources for further studies on conservation and utilization of *L. auriculata* and other *Litsea* species and can lay a solid foundation for understanding the phylogeny of *Litsea*.

## 2. Materials and Methods

### 2.1. Plant Sampling and DNA Extraction of L. auriculata

We sampled fresh young leaves of *L. auriculata* from a cultivated tree at the Shanghai Chenshan Botanical Garden, Shanghai, China (31°4.2609′ N, 121°10.9040′ E), and lodged

a voucher specimen (accession number: TMMJZ20200904) at the herbarium of Nanjing University. Total genomic DNA of *L. auriculata* was extracted from approximately 5 mg of the silica-dried leaf tissue using a modified CTAB method [36]. Subsequently, the integrity of DNA was evaluated by agarose gel electrophoresis and validated using an Agilent 2100 Bioanalyzer (Agilent Technologies, Santa Clara, CA, USA). NanoDrop LITE spectrophotometer (Thermo Fisher Scientific, Wilmington, DE, USA) was employed to measure the concentration of DNA.

### 2.2. Illumina Paired-End Sequencing, De Novo Assembly, and Annotation of the Chloroplast Genome of L. auriculata

The high-quality genomic DNA of *L. auriculata* was used to construct Illumina paired-end (2 × 150 bp) library and implement the low-coverage shotgun sequencing in a lane of HiSeq Xten platform (Illumina, San Diego, CA, USA) at The Beijing Genomics Institute (BGI, Shenzhen, China). Approximately 6 Gb of raw data were sequenced, and the clean data were obtained employing NGS QC Tool Kit by removing adapter sequences and low-quality reads with a Q-value ≤ 20. We de novo assembled the complete chloroplast genome of *L. auriculata* via the GetOrganelle pipeline [37] using the clean data. The chloroplast genome was then automatically annotated using Plastid Genome Annotator [38], with manual adjusting and confirmation of the annotated protein coding genes (CDS) in Geneious Prime (http://www.geneious.com/ (accessed on 10 May 2021), Biomatters Ltd., Auckland, New Zealand). The annotated tRNA genes were further verified using the tRNAscan-SE [39] with default parameters. Afterwards, the resulting annotated cp genome of *L. auriculata* was submitted to The National Center for Biotechnology Information (NCBI; https://www.ncbi.nlm.nih.gov/ (accessed on 11 May 2021)). The circular cp genome physical map of *L. auriculata* was drawn employing the Chloroplot (https://irscope.shinyapps.io/chloroplot/ (accessed on 5 May 2021)) [40], with subsequent manual editing.

### 2.3. Comparative Chloroplast Genome Analysis of Litsea

Chloroplast genome comparison among the 12 *Litsea* species was carried out under the Shuffle-LAGAN mode via mVISTA program (http://genome.lbl.gov/vista/mvista/ (accessed on 8 May 2021)) [41] to elucidate the level of sequence divergence, using the annotation of *L. acutivena* chloroplast genome as a reference.

To further investigate the chloroplast genome-wide evolutionary dynamics and structural variations across these 12 *Litsea* species, we used MAUVE (http://darlinglab.org/mauve/ (accessed on 10 May 2021)) [42] to identify the following evolutionary events: gene loss, duplication, re-arrangements, or translocations in multiple alignments. IRscope (https://irscope.shinyapps.io/irapp/ (accessed on 10 May 2021)) [43] was employed to trace the size variation in the boundary regions of chloroplast genome among inverted repeat regions (IRs), small single copy region (SSC), and large single copy region (LSC).

Additionally, codon usage together with relative synonymous codon usage (RSCU) [44] value was estimated for all protein-coding genes (the genes with sequence length less than 300 bp and repeated genes were eliminated) of 12 whole chloroplast genomes of *Litsea* via CodonW v1.4.2 (http://codonw.sourceforge.net/ (accessed on 8 May 2021)) [45]. The two unique codons (AUG and UGG) and the three stop codons (TAA, TAG, and TGA) have no degeneracy and were deleted from the data before analysis.

### 2.4. Mining of cp Microsatellite Markers and Hypervariable Regions of Litsea

We used the MIcroSAtellite (MISA) perl script [46] to exploit simple sequence repeats (SSRs) within the chloroplast genomes of the studied 12 *Litsea* species, setting the parameters with thresholds of 10 repeat units for mononucleotide SSRs; 6 repeat units for dinucleotide SSRs; and 5 repeat units for tri-, tetra-, penta-, and hexa-nucleotide SSRs. Genus-targeted polymorphic SSRs among these 12 *Litsea* species were selected under the following three criteria: (1) SSRs located in the homologous regions, (2) SSRs that possessed the same repeat units, and (3) the number of repeat units was variable.

Finally, we used DnaSP v6.0 (http://www.ub.edu/dnasp/ (accessed on 8 May 2021)) [47] to calculate the nucleotide variability ($\pi$) of both coding regions and non-coding regions (including intergenic spacers and introns) with aligned length > 200 bp and mutation site > 0 after sequence alignment via MAFFT v7 (https://mafft.cbrc.jp/alignment/software/ (accessed on 10 May 2021)) [48] for identifying the highly variable regions within the chloroplast genomes of 12 *Litsea* species for their future population genetics and species delineation studies. The calculated nucleotide variability values were then plotted in R v4.0.2 (https://www.r-project.org/ (accessed on 2 May 2021)).

### 2.5. Phylogenetic Analysis

To ascertain the phylogenetic position of *Litsea* within the "core Lauraceae" or "core Laureae" group, we employed 12 *Litsea* together with other 50 "core Lauraceae" species representing the *Laurus–Neolitsea*, *Cinnamomum–Ocotea*, and *Machilus–Persea* clades as ingroups and *Caryodaphnopsis tonkinensis* as an outgroup according to the newly updated classification of Lauraceae by Song et al. [6]. Phylogenetic analyses were conducted on the whole chloroplast genome sequences of the above 63 species. The nucleotide sequences were aligned using the default parameters in the MAFFT v7 (https://mafft.cbrc.jp/alignment/software/ (accessed on 10 May 2021)) [48] and followed by some manual adjustments.

We inferred the phylogenetic relationships among the above-studied species using the maximum likelihood (ML) method as implemented in RAxML-HPC v8.2.8 [49] on the CIPRES Science Gateway website (https://www.phylo.org/ (accessed on 11 May 2021)) and the Bayesian inference (BI) method as implemented in MrBayes v3.1.2 [50] under the unpartitioned strategy. A corrected Akaike information criterion (AIC) value was used to determine the best-fitting model of nucleotide substitution and sequence evolution via jModelTest v2.1.10 [51,52], resulting the optimal model of GTR + I + G for both subsequent ML and BI phylogenetic analysis.

Bayesian analyses were conducted using two separated runs of the Markov chain Monte Carlo (MCMC) algorithm for 1 million generations and tree sampling every 1000 generations. The first 25% of sampled trees were discarded as burn-in, and the 25% best-scoring trees were used to construct the consensus tree and to estimate the posterior probabilities (PPs). Convergence was determined by estimating the average standard deviation of the split frequencies (<0.01). To construct the ML tree, we also ran two searches to ensure identical topologies and ML nodal support was calculated with 1000 bootstrap (BS) replicates for each run. Topologies of the above phylogenetic trees were visualized using the Interactive Tree of Life (iTOL) v4 (https://itol.embl.de (accessed on 11 May 2021)) [53].

### 3. Results and Discussion

#### 3.1. Conservation of Litsea Chloroplast Genomes

The Illumina HiSeq Xten platform produced 40,992,320 clean paired-end reads for the chloroplast genome de novo assembly of *L. auriculata*, and the mean sequencing coverage of the chloroplast genome estimated by the GetOrganelle pipeline was 159×. The complete chloroplast genome of *L. auriculata* was 152,377 bp in length and displayed a quadripartite structure consisting of a pair of inverted repeat regions (IR with 20,015 bp) divided by two single-copy regions (LSC, 93,533 bp, and SSC, 18,814 bp; Figure 1, Table 1). The overall GC content of the chloroplast genome was 39.2%, with the corresponding values of 37.9%, 33.9%, and 44.4% for the LSC, SSC, and IR regions, respectively. Moreover, there were a total of 113 unique genes, including 79 protein-coding genes (CDS), 30 transfer RNA (tRNA) genes, and 4 ribosomal RNA (rRNA) genes. Among these genes, 10 protein-coding genes and six tRNA genes contained a single intron, while three protein-coding genes possessed two introns. The gene *rps*12 was trans-spliced: the exon at the 5′ end was located in the LSC region, whereas the 3′ exon and intron were located in the IR regions. Moreover, the $^{\Psi}ycf1$ and $^{\Psi}ycf2$ were identified as pseudogenes because of the partial duplication

(Table 1). The chloroplast genome of *L. auriculata* was deposited in GenBank (NCBI) with the accession number MW355498.

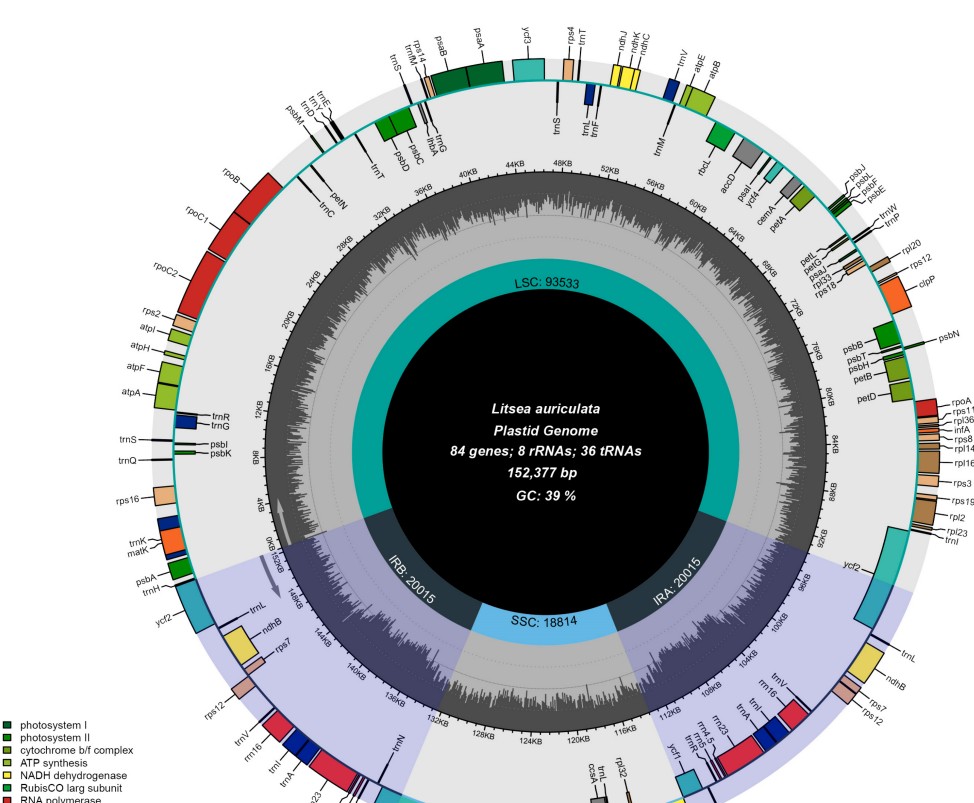

**Figure 1.** Circular map of chloroplast genome of *Litsea auriculata* with annotated genes. Genes shown inside and outside of the circle are transcribed in clockwise and counterclockwise directions, respectively. Genes belonging to different functional groups are color-coded. The GC and AT content are denoted by the dark gray and light gray colors in the inner circle, respectively. LSC, SSC, and IR are large single-copy region, small single-copy region, and inverted repeat region, respectively.

Comparative chloroplast genomics can provide insights into the mechanism of chloroplast evolution of plant species, including the structural rearrangements and gene features [54–57]. In our results, all the 12 *Litsea* chloroplast genomes exhibited the typical and canonical quadripartite structure akin to the majority of angiosperms and other taxa in the Lauraceae [5,6,58]. Variation in chloroplast genome size was very small among 12 *Litsea* species, ranging from 152,132 (*L. szemaois*) to 154,011 bp (*L. garrettii*). The length of their LSC region varied from 93,119 (*L. szemaois*) to 93,827 bp (*L. elongata*), their SSC region from 18,799 (*L. monopetala*) to 18,936 bp (*L. mollis*), and their IR region from 20,015 (*L. auriculata*) to 20,744 bp (*L. garrettii*). The overall GC content was 39.2% across all species except for *L. japonica* and *L. elongata* (39.1%) (Table 1). Gene contents of all 12 chloroplast genomes of *Litsea* species were also highly conserved, and they all encoded an identical set of 113 genes with 30 tRNA genes. Specifically, 15 genes possessed one intron (*atp*F, *ndh*A, *ndh*B, *pet*B, *pet*D, *rpl*2, *rpl*16, *rpo*C1, *rps*16, *trn*A–UGC, *trn*G–UCC, *trn*I–GAU, *trn*K–UUU, *trn*L–UAA, and *trn*V–UAC), and three genes had two introns (*clp*P, *rps*12, and *ycf*3) (Table 2).

We found a high level of similarity in genome-wide organization and evidence of close evolutionary relationships between the 12 *Litsea* species. The chloroplast genome comparisons of the 12 *Litsea* in mVISTA revealed a high sequence similarity between the species (Figure S1) and a lower sequence divergence observed in the IRs than LSC and SSC regions. This probably resulted from the copy correction between IR sequences by gene

conversion [59] and indicated a conserved evolution process of these chloroplast genomes. The non-coding regions possessed higher variation than the coding regions, which was generally consistent with many previous studies on chloroplast genomes [5,6,9,27–29]. Moreover, the MAUVE alignment with the algorithm of progressive Mauve based on the 12 *Litsea* chloroplast genomes showed only one locally collinear block between all analyzed cp genomes, and all of the genes exhibited the same and consistent sequence order, with no gene re-arrangements or inversion events being detected in these genomes (Figure 2).

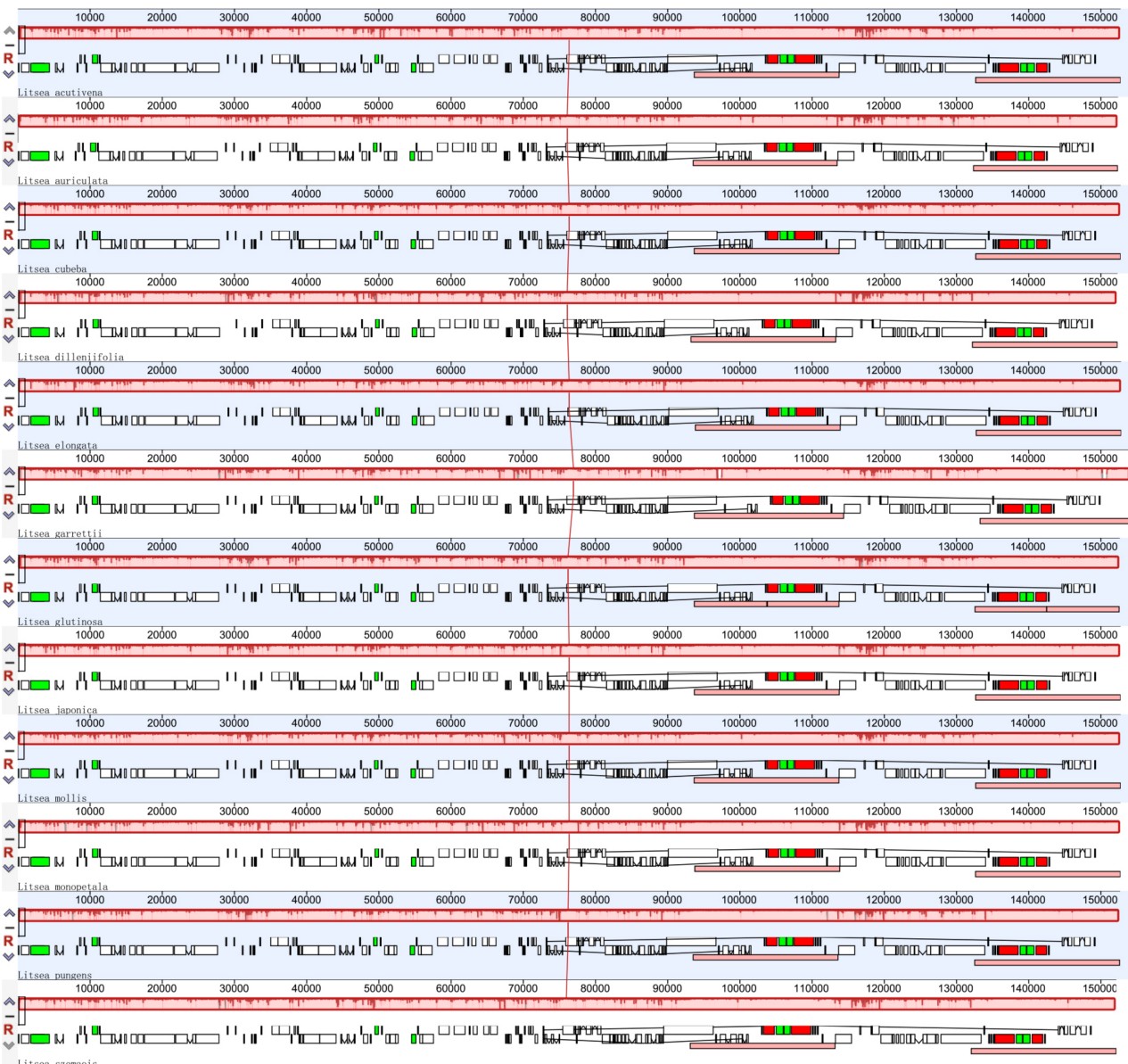

**Figure 2.** Alignment of 12 *Litsea* chloroplast genomes. The *L. acutivena* genome is shown at the top as the reference genome. Within each of the alignment, local collinear blocks are represented by blocks of the same color connected by lines.

**Table 1.** Comparison of complete plastid genomes of 12 *Litsea* species.

| Species | GenBank ID | Whole Sequence Length (bp) | Length of LSC Region (bp) | Length of IR Region (bp) | Length of SSC Region (bp) | Total GC Content (%) | Total Number of Genes | Total Number of CDS Genes | Total Number of tRNA Genes | Total Number of rRNA Genes |
|---|---|---|---|---|---|---|---|---|---|---|
| *L. acutivena* | NC_050362 | 152,718 | 93,677 | 20,066 | 18,909 | 39.2 | 113 | 79 | 30 | 4 |
| *L. auriculata* | MW355498 | 152,377 | 93,533 | 20,015 | 18,814 | 39.2 | 113 | 79 | 30 | 4 |
| *L. cubeba* | NC_048954 | 152,725 | 93,674 | 20,064 | 18,923 | 39.2 | 113 | 79 | 30 | 4 |
| *L. dilleniifolia* | NC_050363 | 152,298 | 93,218 | 20,094 | 18,892 | 39.2 | 112 | 79 | 29 | 4 |
| *L. elongata* | NC_050364 | 152,793 | 93,827 | 20,066 | 18,834 | 39.1 | 113 | 79 | 30 | 4 |
| *L. garrettii* | MN698967 | 154,011 | 93,698 | 20,744 | 18,825 | 39.2 | 113 | 79 | 30 | 4 |
| *L. glutinosa* | KU382356 | 152,618 | 93,690 | 20,061 | 18,806 | 39.2 | 113 | 79 | 30 | 4 |
| *L. japonica* | NC_045267 | 152,718 | 93,697 | 20,066 | 18,889 | 39.1 | 113 | 79 | 30 | 4 |
| *L. mollis* | NC_050366 | 152,736 | 93,655 | 20,063 | 18,936 | 39.2 | 113 | 79 | 30 | 4 |
| *L. pungens* | NC_050368 | 152,655 | 93,520 | 20,131 | 18,873 | 39.2 | 113 | 79 | 30 | 4 |
| *L. szemaois* | NC_050369 | 152,132 | 93,119 | 20,090 | 18,833 | 39.2 | 113 | 79 | 30 | 4 |
| *L.monopetala* | NC_050367 | 152,705 | 93,758 | 20,074 | 18,799 | 39.2 | 113 | 79 | 30 | 4 |

**Table 2.** List of genes in the chloroplast genomes of *Litsea*.

| Groups of Genes | Names of Genes |
|---|---|
| Ribosomal RNAs | *rrn*4.5 (×2), *rrn*5 (×2), *rrn*16 (×2), *rrn*23 (×2) |
| Transfer RNAs | * *trn*A-UGC (×2), ˆ *trn*C-GCA, *trn*D-GUC, *trn*E-UUC, *trn*F-GAA, *trn*G-GCC, * *trn*G-UCC, *trn*H-GUG, *trn*I-CAU, * *trn*I-GAU (×2), * *trn*K-UUU, *trn*L-CAA (×2), * *trn*L-UAA, *trn*L-UAG, *trnf*M-CAU, *trn*M-CAU, *trn*N-GUU (×2), *trn*P-UGG, *trn*Q-UUG, *trn*R-ACG (×2), *trn*R-UCU, *trn*S-GCU, *trn*S-GGA, *trn*S-UGA, *trn*T-GGU, *trn*T-UGU, *trn*V-GAC (×2), * *trn*V-UAC, *trn*W-CCA, *trn*Y-GUA |
| Photosystem I | *psa*A, *psa*B, *psa*C, *psa*I, *psa*J |
| Photosystem II | *psb*A, *psb*B, *psb*C, *psb*D, *psb*E, *psb*F, *psb*H, *psb*I, *psb*J, *psb*K, *psb*L, *psb*M, *psb*N, *psb*T, *psb*Z |
| Cytochrome | *pet*A, * *pet*B, * *pet*D, *pet*G, *pet*L, *pet*N |
| ATP synthase | *atp*A, *atp*B, *atp*E, * *atp*F, *atp*H, *atp*I |
| Rubisco | *rbc*L |
| NADH dehydrogenease | * *ndh*A, * *ndh*B (×2), *ndh*C, *ndh*D, *ndh*E, *ndh*F, *ndh*G, *ndh*H, *ndh*I, *ndh*J, *ndh*K |
| ATP-dependent protease subunit P | ** *clp*P |
| Chloroplast envelop membrane protein | *cem*A |
| Large units | * *rpl*2 (×2), *rpl*14, * *rpl*16, *rpl*20, *rpl*22, *rpl*23, *rpl*32, *rpl*33, *rpl*36 |
| Small units | *rps*2, *rps*3, *rps*4, *rps*7 (×2), *rps*8, *rps*11, ** *rps*12, *rps*14, *rps*15, * *rps*16, *rps*18, *rps*19 |
| RNA polymerase | *rpo*A, *rpo*B, * *rpo*C1, *rpo*C2 |
| Translational initiation factor | *inf*A |
| Miscellaneous proteins | *mat*K, *acc*D, *ccs*A |
| Hypothetical proteins and conserved reading frame | ** *ycf*3, *ycf*4, *ycf*1, *ycf*2 |
| Pseudogene | $^\Psi$*ycf*1, $^\Psi$*ycf*2 |

*Note*: Asterisks (*) before gene names indicate one intron containing genes, and double asterisks (**) indicate two introns in the gene. ×2 indicates genes duplicated in the IR regions. ˆ indicates does not exist in *L. dilleniifolia*. Pseudogene is represented by $^\Psi$.

Additionally, the LSC/IRb (JLB), IRb/SSC (JSB), SSC/IRa (JSA), and IRa/LSC (JLA) boundaries/junctions of the 12 *Litsea* chloroplast genomes (Figure 3) across the 12 *Litsea* species were highly conserved with minor variations. Specifically, the genes *ycf*2, *ndh*F, *ycf*1, *trn*H–GUG, and *psb*A were distributed at the boundaries of LSC/IR and SSC/IR in all 12 *Litsea* species. The entire *ycf*2 crossed the LSC/IRb boundary corresponding to the pseudogene fragment $^\Psi ycf$2 with 3051–3186 bp truncated at the IRa/LSC border, and the gene *ycf*1 was located in the junctions of SSC/IRa among all 12 *Litsea* species, meaning their IRa boundaries all extended into *ycf*1 gene with the extension of the pseudogene fragment $^\Psi ycf$1 ranging from 1371 (*L. mollis*) to 1397 bp (*L. monopetala*) in the IRb/SSC (JSB) boundaries. In addition, the distance between IRb and gene *ndh*F ranged from 16 (*L. garrettii*) to 57 bp (*L. japonica*), and the length between IRa and gene *trn*H–GUG varied from 3 (*L. mollis*) to 23 bp (*L. cubeba*). Correspondingly, the expansion and contraction of the IR regions further explained the length variations in the chloroplast genomes of 12 *Litsea* species, and this phenomenon was quite common in most previously studied angiosperms and the other Lauraceae species [60–62].

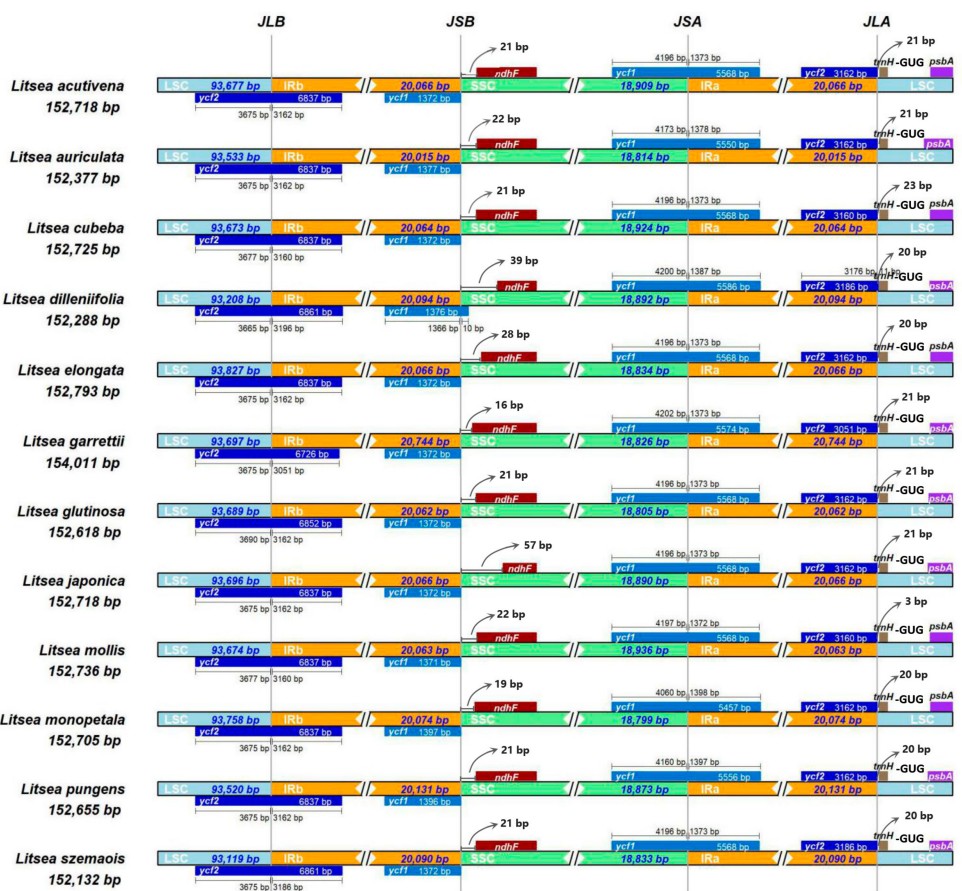

**Figure 3.** Comparison of the borders of the IR, SSC, and LSC regions among 12 chloroplast genomes of *Litsea*. JLB, JSB, JSA, and JLA represent the junctions of LSC/IRb, IRb/SSC, SSC/IRa, and IRa/LSC, respectively.

The comparison results of codon preference indexed by the value of the relative synonymous codon usage (RSCU) showed that 12 *Litsea* species had a common preference for the usage of 30 codons within the 61 shared codons used in the chloroplast genomes (RSCU > 1), and the majority of them were characterized with adenine–thymine ending (Table S1), which was in line with the findings in other Lauraceae plants such as *Cinnamomum bodinieri* Levl. [63], indicating the relative conservation of the cp genes in the genus *Litsea*. However, the preference of the usage of three codons (AUA, GCC, and GGU) varied among the different species of this genus: *Litsea auriculata*, *L. garrettii*, *L. glutinosa*,

*L. monopetala*, *L. pungens*, and *L. szemaois* seldom selected AUA when encoding isoleucine; *L. cubeba*, *L. garrettii*, *L. glutinosa*, and *L. pungens* preferred using GCC than other *Litsea* species; and *L. elongata*, *L. mollis*, and *L. monopetala* had bias on GGU (Table S1). This phenomenon might be explained by the biological characteristics of species or genes formed in the process of long-term evolution and adaptation to the environment and the result of the combined effects of selection, mutation, and drift. Other influencing factors may also include genome size, number of introns, tRNA abundance, and gene expression levels [64,65].

### 3.2. Enrichment of Chloroplast DNA Genetic Resources of Litsea

Microsatellites or SSRs are ubiquitous short tandem repeats often consisting of repetitive sequences/motifs of 1–6 bp in length and can be commonly found in the genomes of diverse organisms [66]. Owing to high level of polymorphisms, reproducibility, and abundance in plant genomes, they are widely used as one of the most important and valuable molecular markers for plant population genetic analysis, evolutionary studies, and molecular marker-assisted selection in plant breeding programs [58,67]. Here, we detected a total of 876 chloroplast SSRs within the chloroplast genomes of the 12 *Litsea* species by MISA, with the number of chloroplast SSRs ranging from 67 (*L. mollis*) to 80 (*L. auriculata*). The majority of these chloroplast SSRs (548 out of 876 or 62.56%) were mononucleotides composed mainly of short polyadenine (polyA) or polythymine (polyT) repeats and much less often contained guanine (G) or cytosine (C) tandem repeats. The remaining chloroplast SSRs were comprised of complex nucleotide repeats (13.81%), dinucleotides (11.99%), tetranucleotides (9.25%), trinucleotides (1.83%), pentanucleotides (0.34%), and hexanucleotides (0.11%) (Table 3).

**Table 3.** Summary of the simple sequence repeats (SSR) in 12 *Litsea* species.

| Species | SSR Numbers | P1 Loci (N) | P2 Loci (N) | P3 Loci (N) | P4 Loci (N) | P5 Loci (N) | P6 Loci (N) | Pc Loci (N) | LSC | SSC | IR |
|---|---|---|---|---|---|---|---|---|---|---|---|
| *L. acutivena* | 72 | 46 | 9 | 1 | 8 | – | – | 8 | 55 | 13 | 4 |
| *L. auriculata* | 80 | 55 | 9 | 1 | 5 | – | – | 10 | 62 | 14 | 4 |
| *L. cubeba* | 71 | 43 | 9 | 1 | 8 | – | 1 | 9 | 55 | 12 | 4 |
| *L. dilleniifolia* | 71 | 42 | 9 | 2 | 8 | – | – | 10 | 55 | 12 | 4 |
| *L. elongata* | 74 | 46 | 8 | 2 | 6 | – | – | 12 | 58 | 12 | 4 |
| *L. garrettii* | 75 | 46 | 9 | 1 | 7 | – | – | 12 | 58 | 13 | 4 |
| *L. glutinosa* | 72 | 45 | 11 | 1 | 7 | 1 | – | 7 | 57 | 9 | 6 |
| *L. japonica* | 72 | 44 | 7 | 1 | 6 | – | – | 14 | 56 | 12 | 4 |
| *L. mollis* | 67 | 41 | 8 | 1 | 7 | – | 1 | 9 | 53 | 10 | 4 |
| *L. pungens* | 73 | 48 | 9 | 2 | 6 | – | – | 8 | 58 | 11 | 4 |
| *L. szemaois* | 69 | 43 | 8 | 2 | 7 | – | – | 9 | 54 | 11 | 4 |
| *L. monopetala* | 80 | 49 | 9 | 1 | 6 | – | 2 | 13 | 58 | 16 | 6 |
| Total/ Percentage | 876 (100%) | 548 (62.56%) | 105 (11.99%) | 16 (1.83%) | 81 (9.25%) | 1 (0.11%) | 4 (0.46%) | 121 (13.81%) | 679 (77.51%) | 145 (16.55%) | 52 (5.94%) |

*Note*: P1–P6 represent SSRs of the mononucleotide, dinucleotide, trinucleotide, tetranucleotide, pentanucleotide, and hexanucleotide types, respectively. Pc represents a complex nucleotide repeat. LSC and SSC denote large and small single copy regions, respectively. The last row represents the percentage for each type of SSR. "–" indicates no data.

Consistent with many previous reports [6,9], the chloroplast SSRs in *Litsea* were mainly located in the LSC regions (77.51%). A smaller percentage of the chloroplast SSRs occurred in the SSC (16.55%) and IR (5.94%) regions, respectively. These chloroplast SSRs were also richer in the non-coding regions (84.7%), such as intergenic spacers and introns, than coding regions (15.3%). The complete detail of repeat types and locations of cpSSRs in each *Litsea* species are listed in Table S2. Moreover, chloroplast SSRs exhibited high diversity and variations in copy numbers [56,68], and thus we identified 10 (6 mononucleotides, 3 trinucleotides, 1 tetranucleotides) polymorphic chloroplast SSRs within the 12 *Litsea* species (Table S2). These chloroplast SSR molecular markers developed in our study will be useful in population genetics and evolutionary studies of the genus *Litsea* as well as the molecular breeding and conservation of this and related genera.

Chloroplast DNA molecular markers have been extensively used for research on plant population genetics, phylogeny, phylogeography, and DNA barcodes for species identification and delimitation [67,69,70] by virtue of their advantages of low rates of nucleotide substitutions, usually uniparental inheritance and recombination [58,70]. On the basis of our comparative chloroplast genomics results, we found that the average nucleotide diversity in the intergenic spacer regions (mean $\pi$ = 0.00835) was significantly higher than that in the protein coding gene regions (mean $\pi$ = 0.00326) and introns (mean $\pi$ = 0.00305) (Figure 4, Table S3). For the intergenic spacer (IGS) and intron regions, $\pi$ values varied from 0.0006 (IGS, *trn*L–*ndh*B) to 0.05523 (IGS, *psb*E–*pet*L), and the top 10 hypervariable regions were *psb*E–*pet*L, *trn*H–*psb*A, *pet*A–*psb*J, *ndh*F–*rpl*32, *ycf*4–*cem*A, *rpl*32–*trn*L, *ndh*G–*ndh*I, *psb*C–*trn*S, *trn*E–*trn*T, and *psb*M–*trn*D ($\pi$ > 0.0094) (Figure 4, Table S3). Regarding the protein coding gene regions, pairwise sequence divergence values ($\pi$) for each region ranged from 0.00021 (*ndh*B) to 0.00832 (*ndh*F), whereas five regions (*ndh*D, *mat*K, *ccs*A, *ycf*1, *ndh*F) had remarkably high values ($\pi$ > 0.005). Compared with other previous DNA barcoding studies of Lauraceae, for instance, the fragments of *rpl*32–*trn*L, *ndh*F–*rpl*32, and *ycf*1 had also been identified as variable regions in the tribes Laureae, Cinnamoneae, and Perseeae [28,63,71–73]. The *rpl*32–*trn*L region could also be used to further distinguish *Alseodaphne*, *Alseodaphnopsis*, and *Dehaasia* [72–75], while the *psb*A–*trn*H, *mat*K, and *psb*C–*trn*S had proven to be effective for resolving the taxonomy and phylogeny of *Machilus* and *Phoebe* also [5,59,60]. In our study, the fragments *ndh*D, *mat*K, *ccs*A, *ycf*1, *ndh*F, *psb*E–*pet*L, *trn*H–*psb*A, *pet*A–*psb*J, *ndh*F–*rpl*32, *ycf*4–*cem*A, *rpl*32–*trn*L, *ndh*G–*ndh*I, *psb*C–*trn*S, *trn*E–*trn*T, and *psb*M–*trn*D were found to be particularly variable loci among *Litsea* chloroplast genomes, which could be employed for DNA barcoding or intraspecific delimitations, as well as for facilitating a better-resolved molecular phylogeny of *Litsea* species.

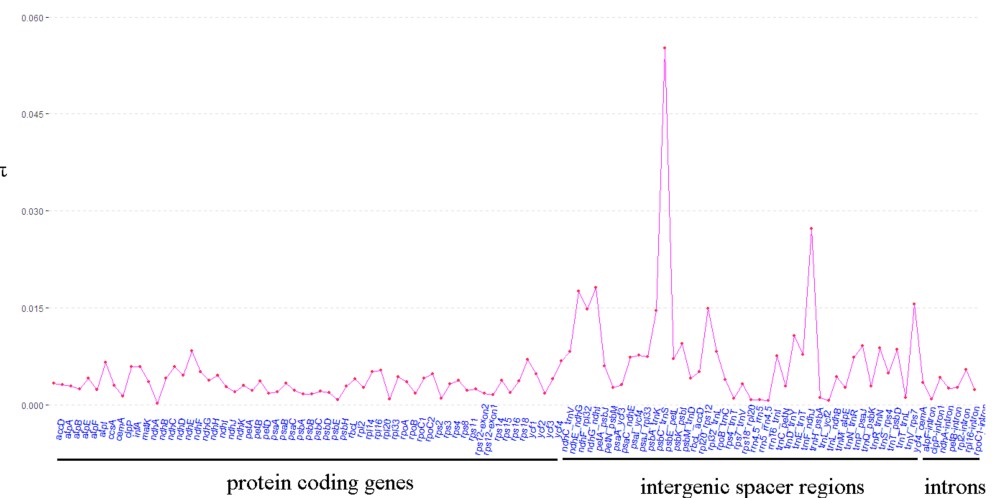

**Figure 4.** Nucleotide variability ($\pi$) values of 12 *Litsea* chloroplast genomes.

*3.3. Phylogenetic Analysis*

The phylogenetic trees resulting from the maximum likelihood and Bayesian inference analyses yielded identical tree topologies and showed that the studied *Litsea* species can be divided into the following four sub-clades on the basis of the node supports (Figure 5). Sub-clade I (BS = 100, PP = 1.00) contained two sibling *Litsea* species (*L. mollis* and *L. cubeba*) and *Lindera obtusiloba*. The sub-clade II, which contained *Actinodaphne lancifolia* (*L. coreana*), *L. auriculata*, *L. szemaois*, *L. dilleniifolia*, *L. monopetala*, *L. garrettii*, *L. elongata*, and *L. japonica*, was sister to sub-clade I. Within sub-clade II, the grouping of *L. auriculata*, as well as its sister taxon *Actinodaphne lancifolia*, was well supported with a of BS value 80 and PP of 1.00 (Figure 5). Six *Litsea* species (*L. szemaois*, *L. dilleniifolia*, *L. monopetala*, *L. garrettii*, *L. elongata*, *L. japonica*) formed a cluster into within sub-clade II, which was sister to the *L. auriculata–A. lancifolia* pair. The other two *Litsea* species (*L. glutinosa*, *L. acutivena*), *Lindera megaphylla*, and *Laurus nobilis* formed sub-clade III with full support of BS 100 and PP 1.00,

within which *Litsea acutivena* was sister to *Lindera megaphylla* (Figure 5). The sub-clade IV was sister to sub-clade III and contained the highly supported sister taxa of *Litsea pungens* and *Lindera floribunda* (with a BS of 100 and PP of 1.00), which were sister taxa to five other *Lindera* taxa (*L. rubronervia, L. praecox, L. neesiana, L. sericea, L. reflexa*) (Figure 5).

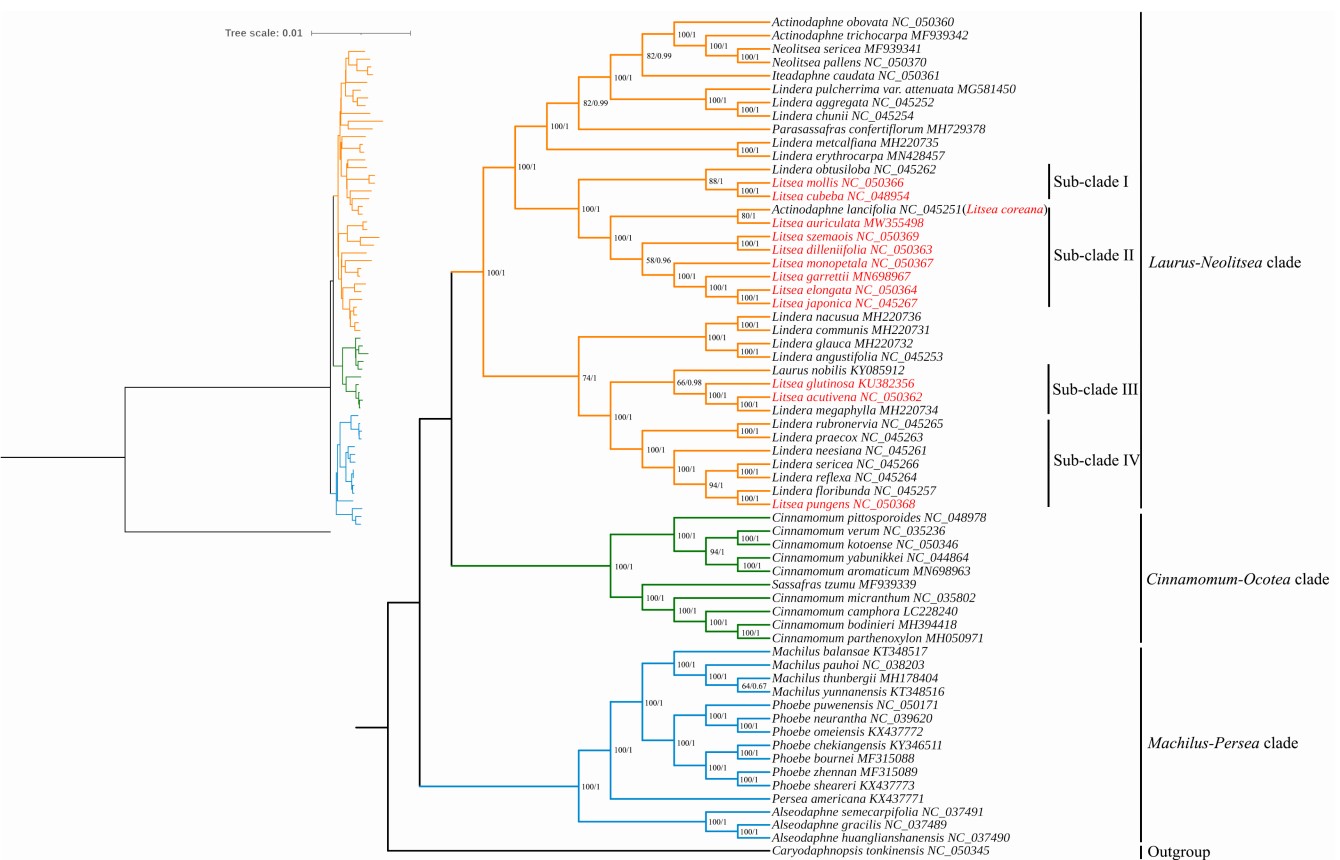

**Figure 5.** Complete chloroplast genome phylogenetic tree of 63 species inferred from maximum likelihood (ML) and Bayesian inference (BI) analysis. Numbers above the lines represent ML bootstrap values (BS) and BI posterior probabilities (PP).

The new chloroplast genome phylogeny of *Litsea* enables us to elucidate the phylogenetic relationships among *Litsea* and related genera. First and foremost, the phylogenetic relationships between *Litsea*, *Lindera*, and *Laurus* have long been complex and controversial [3,76]. All three genera share similar morphological traits such as having umbellate inflorescences subtended by large involucral bracts, introrse anthers, the presence of two stipitate glands at the third whorl, and oval or spherical fruits [3,76]. Previously, *Lindera* and *Litsea* were differentiated on the basis of the number of anther cells [21], but our tree topology and recent work by Tian et al. [28] showed that this character may not be as taxonomically useful as previously thought for differentiating these genera. It is likely that future work with more comprehensive sampling of *Litsea* and *Lindera* will reveal the extend of the polyphyly of both genera.

Our phylogenetic tree also revealed the surprising relationship between *Litsea auriculata* and *Actinodaphne lancifolia*. A deeper look at the nomenclature of *Actinodaphne lancifolia* revealed that *Litsea coreana* H. Lév. is a synonym of this species [77]. The sessile umbels, persistent bracts, and scattered leaves of *A. lancifolia* in conjunction with our molecular data supports the reinstatement of *Litsea coreana* Lévl. as the correct name for the taxon. Additionally, our phylogenetic trees demonstrated full support (BS = 100, PP = 1.00) that the genus *Actinodaphne* was most related to the genus *Neolitsea*, specifically that *Actinodaphne trichocarpa* was sister to *Neolitsea sericea* and *Neolitsea pallens*, and *Actinodaphne obovata* was closely related with them, which is consistent with the research of Song et al. [6].

To sum up, our phylogenetic study resolved here revealed that the *Litsea* group (among the 12 species investigated) containing in the *Laurus–Neolitsea* clade was polyphyletic and closely related to the genera *Lindera* and *Laurus*. Moreover, our ML and BI analyses with complete plastid genomes grouped all ingroup taxa into three clades (*Laurus–Neolitsea* clade, *Cinnamomum–Ocotea* clade, *Machilus–Persea* clade), and our phylogenetic topologies further supported the monophyly of the above three clades, which was also in accordance with the previously comprehensive phylogenetic tree constructed by Song et al. [6] on the basis of the cp genome data. Therefore, whole plastome sequencing has been proven to display higher resolution and reliability in resolving phylogenetic relationships at a wide range of taxonomic levels. On the other hand, to illuminate the more transparent systematic placements of *Litsea* and its related genera, we recommend more intensive and comprehensive taxa sampling and the combination of other sources of molecular data such as the transcriptomes and whole genomes [78,79]. Considering hybridization, gene introgression, polyploidization, and stochastic and systematic errors may also complicate the use of molecular data as the taxonomic proof of phylogeny [72–75], and thus we propose an integrated analysis of phenotypic traits such as inflorescence characters and molecular data for better resolving and clarifying the Lauraceae systematic research.

## 4. Conclusions

The chloroplast genome of *Litsea auriculata* was determined and charactered for the first time in this study, and we conducted the comparative plastid genomics and phylogenomics with that of 11 other *Litsea* species and 51 "core Lauraceae" species. Our results showed the high conservativeness of the plastid genomes of these 12 *Litsea* species regarding the canonical angiosperm quadripartite structure with no structural arrangement or gene inversion and the unbiased codon usage preferences. In the meantime, abundant genetic resources including 10 intergenic spacers (*psb*E–*pet*L, *trn*H–*psb*A, *pet*A–*psb*J, *ndh*F–*rpl*32, *ycf*4–*cem*A, *rpl*32–*trn*L, *ndh*G–*ndh*I, *psb*C–*trn*S, *trn*E–*trn*T, and *psb*M–*trn*D), 5 protein coding genes (*ndh*D, *mat*K, *ccs*A, *ycf*1, and *ndh*F), and 876 cpSSRs were developed here as potential DNA barcodes and variable molecular markers for further research of delimitation, phylogenetics, population genetics, and evolution of the genus *Litsea*, as well as the molecular breeding and conservation of this and the related genus. Our chloroplast genome phylogeny revealed that the 12 *Litsea* species investigated were polyphyletic, closely related to the genera *Lindera* and *Laurus*, and nested within the *Laurus–Neolitsea* clade. In addition, our analyses with complete chloroplast genomes grouped all ingroup taxa into three clades (*Laurus–Neolitsea* clade, *Cinnamomum–Ocotea* clade, *Machilus–Persea* clade), and our phylogenetic topologies further supported the monophyly of the above three clades previously delimited by Song et al. [6]. We conclude that whole chloroplast genome sequencing enables the development of useful molecular markers and allows for higher resolution and reliability in resolving phylogenetic relationships of the Lauraceae at a wide range of taxonomic levels.

**Supplementary Materials:** The following are available online at https://www.mdpi.com/article/10.3390/f12060744/s1, Figure S1: Comparison of the cp genomes among the 12 *Litsea* species via mVISTA using annotation of *L. acutivena* as a reference. Table S1: RSCU analysis of protein-coding regions in 12 *Litsea* species. Table S2: The identified cpSSR loci and genus-targeted polymorphic cpSSRs in 12 *Litsea* species. Table S3: The nucleotide variability value (π) of both coding regions and non-coding regions (including intergenic spacers and introns) in 12 *Litsea* species.

**Author Contributions:** Conceptualization, P.L. and Z.W. (Zhongsheng Wang); methodology, Y.Z. (Yunyan Zhang), Y.T., and Y.Z. (Yuntian Zhang); software, Y.T., Y.Z. (Yunyan Zhang), D.Y.P.T., and Y.Z. (Yuntian Zhang); validation, Y.Z. (Yunyan Zhang), D.Y.P.T., and Z.W.; formal analysis, Z.W. (Zhengwei Wang), Y.Z. (Yunyan Zhang), and J.Z.; investigation, Y.Z. (Yuntian Zhang) and J.Z.; resources, P.L. and Z.W. (Zhongsheng Wang); data curation, Y.T., Y.Z. (Yuntian Zhang), and D.Y.P.T.; writing—original draft preparation, Y.Z. (Yunyan Zhang) and Y.T.; writing—review and editing, D.Y.P.T., P.L., and Z.W. (Zhengwei Wang); visualization, Y.Z. (Yunyan Zhang), Y.T., Y.Z. (Yuntian Zhang), and J.Z.; supervision, Z.W. (Zhengwei Wang), Y.Z. (Yuntian Zhang), and J.Z.; project administration, P.L. and

Z.W. (Zhengwei Wang); funding acquisition, P.L. and Z.W. (Zhongsheng Wang) All authors have read and agreed to the published version of the manuscript.

**Funding:** This research was funded by the National Natural Science Foundation of China (grant no. 30970512) and the State Key Basic Research and Development Plan of China (grant no. 2017YFA0605104).

**Data Availability Statement:** The data presented in this study are available in the article and Supplementary Materials. The whole chloroplast genomes data of this study are openly available in GenBank of NCBI at https://www.ncbi.nlm.nih.gov; accession number: MW355498 (accessed on 11 May 2021).

**Acknowledgments:** The authors sincerely thank Mengyuan Zhang for his great help in collecting plant materials, Ruisen Lu for his kind instructions on drawing high-quality pictures, and Shan Lu for his helpful instruction for preserving the plant specimen at the herbarium of Nanjing University.

**Conflicts of Interest:** The authors declare no conflict of interest.

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
