# Peer review of "Comparative Chloroplast Genomics of Litsea Lam. (Lauraceae) and Its Phylogenetic Implications"

_forests, doi:10.3390/f12060744_

Round 1
Reviewer 1 Report
Broad comments:
This work is quite well designed and written. The subject and scientific problem is also intresting.
The introduction introduces the reader well to the issues related to Litsea Lam. Which belongs to the “core Lauraceae” or “core Laureae” both in terms of systematics and the need for protection. The introduction also details the results of previously conducted genetic research. From the methodological point of view this MS is also well done. All analysis parameters are described in sufficient detail. Presentation of obtained results is also very good and clear.
The conclusions are quite general in nature, but they are correctly formulated and are confirmed by the results obtained.
Specific comments:
Please consider whether it would not be good to outline the purpose of the research more clearly in the introduction.
Line 142: In the sentence "develop genetic markers such as microsatellites and DNA barcodes for ... maybe it is worth adding potentially? Because it has not been fully checked if they really work. These are only some suggestions.
Line 218: Phylogenetic analyzes were conducted on the whole chloroplast genome sequences of the above 63 species. Why exactly 63? In my opinion, probably too much, because Figure 5 is not readable.
Reviewer 2 Report
Comments and Suggestions for Authors
The article is focused on describing fully sequenced Litsea auriculate chloroplast genome and phylogenetic reconstruction of the genus Litsea. Studied species is a representative of genus, which consists of many endangered species also important in Chinese natural medicine culture. Therefore, the research task seems to be important and well justified.
The introduction familiarizes reader with the topic well and highlights why the research of endangered species is important to the scientific community. Authors engage well with other literature in the field. In my opinion authors have chosen the best approach to this study. Bioinformatic tools as well as sequencing method are widely used and accepted by the scientists specialized in molecular taxonomy. The results and the discussion exhaustively describe the obtained results and the current state of knowledge and do not cause any logical doubts. They also provide information on new potential markers for molecular taxonomy research. Conclusions drawn based upon the data presented are valid and reasonable. Research brings new important information to the general knowledge of biodiversity. Article presents high-quality research that brings new information to understanding biodiversity of the Litsea genus. Those features I consider as strengths of this article. The manuscript does not need any improvement of the merits.
My only concerns are rather technical. Figure 5 seems to me to be difficult to read. Please take into account whether it could be enlarged. The text should only be double-checked for typos and punctuation marks.
I recommend publication of this article after minor revision of above mentioned typos, punctuation marks and Figure 5.
